# 🦖 DYNOSAUR: A Dynamic Growth Paradigm for Instruction-Tuning Data Curation

**Da Yin**[*][§]    **Xiao Liu**[*][♣]    **Fan Yin**[*][§]    **Ming Zhong**[*][†]

**Hritik Bansal**[§]    **Jiawei Han**[†]    **Kai-Wei Chang**[§]

[§]UCLA    [♣]Peking University    [†]UIUC

{da.yin, fanyin20, hbansal, kwchang}@cs.ucla.edu    lxlisa@pku.edu.cn

{mingz5, hanj}@illinois.edu

dynosaur-it.github.io

## Abstract

Instruction tuning has emerged to enhance the capabilities of large language models (LLMs) to comprehend instructions and generate appropriate responses. Existing methods either manually annotate or employ LLM (e.g., GPT-series) to generate data for instruction tuning. However, they often overlook associating instructions with existing annotated datasets. In this paper, we propose DYNOSAUR, a dynamic growth paradigm for the automatic curation of instruction-tuning data. Based on the metadata of existing datasets, we use LLMs to automatically construct instruction-tuning data by identifying relevant data fields and generating appropriate instructions.

By leveraging the existing annotated datasets, DYNOSAUR offers several advantages: 1) it reduces the API cost for generating instructions (e.g., it costs less than $12 USD by calling `GPT-3.5-turbo` for generating 800K instruction tuning samples; 2) it provides high-quality data for instruction tuning (e.g., it performs better than ALPACA and FLAN on SUPER-NI and LONGFORM with comparable data sizes); and 3) it supports the continuous improvement of models by generating instruction-tuning data when a new annotated dataset becomes available. We further investigate a continual learning scheme for learning with the ever-growing instruction-tuning dataset, and demonstrate that replaying tasks with diverse instruction embeddings not only helps mitigate forgetting issues but generalizes to unseen tasks better.

Code and data are available at https://github.com/WadeYin9712/Dynosaur.

## 1 Introduction

Instruction tuning (Sanh et al., 2022; Ouyang et al., 2022; Wei et al., 2022) enables large language models (LLMs) (Raffel et al., 2020; Brown et al., 2020; Touvron et al., 2023) to provide appropri-

ate output according to input instructions. Existing approaches compile instruction-tuning datasets mainly by 1) manual annotations or 2) distillate from a larger size of LLM. For example, SUPER-NATURALINSTRUCTION (SUPER-NI) (Wang et al., 2022b) and DOLLY (Databricks, 2023) recruit experts to manually annotate task instructions and related task data. Despite their high quality, this approach is labor-intensive and costly (Honovich et al., 2022a). Recent efforts (Wang et al., 2022a; Taori et al., 2023) leverage GPT-series to distill instruction tuning data to train smaller models. However, subsequent studies (Gudibande et al., 2023) argue that these methods merely help smaller models learn to mimic the style of teacher LLMs without inheriting their true capabilities, such as factuality and problem solving skills. We suspect it is mainly due to the instructions not ground to actual data.

In this paper, we propose 🦖 DYNOSAUR, a dynamic growth paradigm to convert high quality annotations from dataset repositories into instruction-tuning data. In particular, DYNOSAUR generates instructions based on the metadata of existing datasets in the dynamically growing Huggingface Datasets Platform (Lhoest et al., 2021). As shown in Figure 1, metadata covers essential information about a dataset, including dataset description ("A collection of ... ebooks ..."), dataset name ("Gutenburg_English"), data fields ("title", "text", ..., "issued") and dataset annotations. Guided by metadata, our method can generate multiple tasks applicable for forming instruction-tuning data with instances in NLP datasets. We leverage LLMs to harvest task instructions and their corresponding input/output fields with a *single* prompt. Prompted with dataset description involving ebooks and data fields about the book published information, LLMs can synthesize instructions such as "Given a Gutenburg passage, generate its title" and "Predict the year when the book is published based

---

[*]These four authors contributed equally.

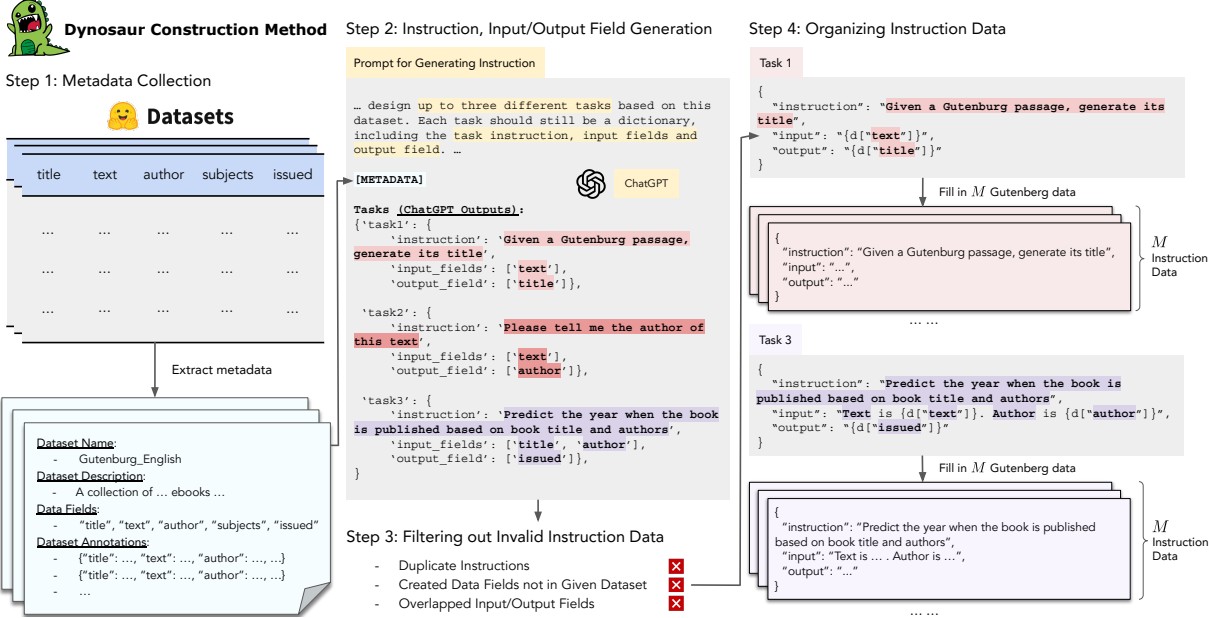

Figure 1: Overall pipeline of collecting DYNOSAUR data. "d" in Step 4 means each instance in Gutenberg dataset.

on book `title` and `authors`". These instructions reflect the original data domain and use multiple dataset components.

In the meantime, LLMs also determine which data fields should be used to construct corresponding task inputs/outputs according to generated instructions. As illustrated in Figure 1, LLMs capture corresponding input fields "`title`" and "`author`" and output field "`issued`" for the generated task about predicting issued years given book title and authors. Subsequently, all the data under "`title`" and "`author`" fields are used as the final inputs of the generated task, and the data under "`issued`" are treated as final outputs. Suppose that we generate $N$ instructions based on the metadata of a dataset which contains $M$ instances, our method can synthesize $N \times M$ instruction-tuning data.

DYNOSAUR offers several advantages:

**Low Conversion Cost.** As DYNOSAUR leverages existing annotated data, it reduces the number of queries to larger LLMs for generating instructions. For example, it costs only $11.5 USD to query GPT-3.5-turbo (OpenAI, 2023) and generate 800K instruction-tuning data based on annotated datasets. In contrast, both ALPACA and INSTRUCTION GPT-4 cost around $500 USD to generate a significantly smaller dataset of 52K instances. Despite the lower cost of querying LLMs, DYNOSAUR generates high-quality data by effectively leveraging existing annotations.

**Effectiveness of Instruction-Tuning Data.** We evaluate the data effectiveness by studying whether models trained with DYNOSAUR can achieve competitive performance on SUPER-NI, LONGFORM (Köksal et al., 2023) and USER-INSTRUCTION-252 (Wang et al., 2022a). On SUPER-NI, both T5-3B and LLAMA-7B models fine-tuned with DYNOSAUR outperform ALPACA, INSTRUCTION GPT-4 and DOLLY that are much more expensive to be collected. In particular, training T5-3B with DYNOSAUR brings 2.5-22 ROUGE-L improvement than other datasets. On LONGFORM, training T5-3B with DYNOSAUR is 2.8-12.8 METEOR better than training with other human-curated instruction data such as PROMPT-SOURCE (Sanh et al., 2022) and FLAN (Wei et al., 2022). On USER-INSTRUCTION-252, DYNOSAUR can be exploited as additional training data to achieve higher performance than solely training with either ALPACA or INSTRUCTION GPT-4.

**Supporting Continuously Improving Models with New Instruction Data.** Statistics show that an average of 143.6 datasets are added to Huggingface Datasets daily in 2023. Because of the low conversion cost, DYNOSAUR can grow dynamically as the platform expands without much effort.

An ever-growing instruction-tuning dataset provides an opportunity to continuously improve instruction-following models. Suppose we have a model trained with $K$ tasks ($\mathcal{M}_K$) and newly obtain $L$ training tasks. How can we train $\mathcal{M}_K$ with

the $L$ new tasks to 1) achieve better generalization on unseen tasks and the new $L$ tasks and 2) suffer less from forgetting the previous $K$ training tasks? We propose several continual learning strategies specifically for instruction tuning which select replay tasks based on the diversity of instruction and data representations. Experiments with SUPER-NI and DYNOSAUR show that replaying is effective to improve generalization and mitigate forgetting. Besides, once $L$ new tasks are used for training, replaying previous tasks with the least similar instructions to the $L$ tasks performs the best.

## 2 Collection of DYNOSAUR Data

In this section, we introduce how to construct the DYNOSAUR dataset. As shown in Figure 1, we first collect metadata from existing datasets, then prompt LLM to create tasks based on the metadata, and filter out invalid ones.

### 2.1 Metadata Collection

Metadata contains key information about an NLP dataset that contributes to instruction-tuning data generation. It covers the following elements:

**Dataset Name.** Dataset name sometimes provides useful information to help us identify the domain and task category of a dataset. For example, dataset names with "bio" usually indicate that the dataset is in the biological domain; names with "nli" may suggest that the dataset is originally designed for natural language inference tasks.

**Dataset Description.** Dataset description offers more detailed information about the motivation for building a dataset, the summary of dataset contents, and its supported tasks. It facilitates LLM to create instructions by supplying extra information about the dataset domain and initial dataset design.

**Data Fields and Dataset Annotations.** Data fields are the keys included in dataset annotations. For example, given an instance {"title": ..., "text": ..., "author": ..., "subjects": ..., "issued": ...}, the data fields are "title", "text", "author", "subjects" and "issued". When LLM generates task instructions, it needs to determine which fields can be used as task inputs/outputs according to the semantics of data field names and contents of the data fields.

All the metadata components are collected from the Huggingface Datasets Platform. We only col-

lect the metadata from datasets whose licenses allow adaptation. More details are in Appendix A.

### 2.2 Instruction and Input/Output Field Generation

For each dataset accompanied by processed metadata, we then deploy LLM to generate multiple tasks associated with it. For each task, LLM generates a specific task instruction and designates its input/output fields simultaneously. As exemplified in Figure 1, LLM is expected to generate an instruction "Given a Gutenburg passage, generate its title", its input field "text", and the output field "title".

To accomplish this, we harness the power of in-context learning (Brown et al., 2020). Concretely, we wrap the information of each dataset into a dictionary format and construct four demonstrations manually. Due to the length limitation of the LLM, we use two of them each time as part of the input. Depending on whether or not the incorporating dataset descriptions in the input prompt, we consider the following two configurations:

**Description-Aware Generation.** To maximize the utilization of information present in the dataset description, we incorporate metadata of the two demonstration datasets as well as the new dataset where we plan to generate new tasks as input. The benefit is that LLM can infer the underlying purpose of the dataset creation, thereby generating the most aligned tasks with the original intent. In this setup, LLM generates new tasks, with the input prompt being: "Now given a dictionary as input, please help us to generate new tasks. You may stop when there is no more plausible task." and requirements being "Note that the input and output fields should not be duplicated and should both appear in [data fields]. Each task should still be a dictionary, containing no text or explanations outside the dictionary." The full prompt is shown in Appendix B. This setting, however, still has limitations: firstly, comprehensive metadata may not be available for certain datasets; secondly, LLM exhibits a proclivity towards dataset descriptions, leading to homogenization of the generated tasks. To mitigate these issues, we additionally introduce the following setup.

**Description-Unaware Generation.** To fully exploit the annotations and distinct data fields, we exclude the dataset description from the

input, thereby allowing the LLM to freely generate diverse task instructions and input/output fields. In this scenario, the dataset can be perceived as a description-less database, with the LLM generating diverse potential tasks based on the valid fields within it. For instance, the data fields in the Wikipedia-based QA dataset may encompass "title", "context", "question", and "answers". Possible new tasks could include Wikipedia article generation ("title"⇒"context"), Wikipedia title generation ("context"⇒"title"), and open-domain QA question generation ("answer"⇒"question").

By integrating these two settings, we ensure the preservation of the original intent of all datasets, while leveraging the creativity of LLM to delve deeper into the inherent potential in existing data.

## 2.3 Post-Processing

**Filtering Invalid Tasks.** Even though we describe the requirements for a valid task in the prompt, LLM sometimes neglects the requirements and generate invalid tasks. We filter out tasks with three criteria: 1) tasks with non-existent data fields (for instance, a task with the output field "content" is invalid given the data in Figure 1); 2) tasks with more than one output fields; 3) tasks whose input/output fields overlap. Moreover, we remove duplicate tasks created during both the description-aware and -unaware generation.

**Organizing Instruction Data.** We organize the instruction data in the form of "instruction", "input", and "output". Given an instance of a dataset and a generated task containing the instruction, input fields, and the output field, the "instruction" is the generated instruction and the "output" is the value of the output field. If there is only one input field, the "input" is the value of the input field; otherwise, the "input" describes all the input fields with the format "The [field name] is [value of the field]."

**Adding Label Spaces for Classification Tasks.** As we only showcase several dataset instances to LLMs, it does not know the entire label space when generating a classification task. As a result, the generated instruction may not contain the label space knowledge adequately. To overcome this issue, we automatically add the label space information in the instruction of classification tasks. We simply treat a task with less than 10 distinct outputs as a classification task, and add "Answers must be one of [distinct outputs]." to the end of the instruction. We also discard classification tasks with extremely imbalanced distributions (e.g., only one distinct output value) in this step.

## 2.4 Statistics and Cases

In total, we collect 2,911 English datasets from the Huggingface Datasets Platform as of Feb 23, 2023. We then feed them to GPT-3.5-turbo (OpenAI, 2023) and generate 13,610 tasks, of which 5,740 are valid and distinct. For each task, we sample up to 200 instances, ending in 801,900 instances that form the DYNOSAUR dataset. The diversity of the instructions is shown in Figure 3. Following the approach of Wang et al. (2022a), we plot the top 20 most prevalent root verbs and their top 4 direct nouns, each of which appears at least 5 times. The instructions are quite diverse, especially when considering we only have a total of 4 demonstrations.

Figure 2 demonstrates examples of datasets and corresponding tasks. The dataset name, dataset description, data fields, and annotations are all used by LLM to design the tasks. LLM infers from the dataset name that it is about anaphor agreement and include this information in the instruction. In Example 2, LLM creates the task of paraphrase identification by understanding the relationship between the fields "sentence1" and "sentence2" implied in the dataset description. Under the description-unaware setting like Example 3, tasks can be generated based on the names of data fields.

## 3 Experiments

We conduct two sets of experiments to evaluate the quality of DYNOSAUR. We first evaluate models trained with DYNOSAUR on SUPER-NI and LONGFORM to examine its ability to solve NLP tasks. Then we run a human evaluation to examine if DYNOSAUR helps in user-oriented situations.

### 3.1 Automatic Evaluation on SUPER-NI and LONGFORM

**Experimental Settings.** We fine-tune T5-3B and LLAMA-7B with a variety of instruction-tuning datasets, including DYNOSAUR, SUPER-NI training set, ALPACA, etc. LLAMA-7B is fine-tuned with LORA (Hu et al., 2022), an efficient fine-tuning approach. We also compare with larger models, including models based on T5-11B, T0 and T0++ (Sanh et al., 2022), Tk-Instruct (Wang

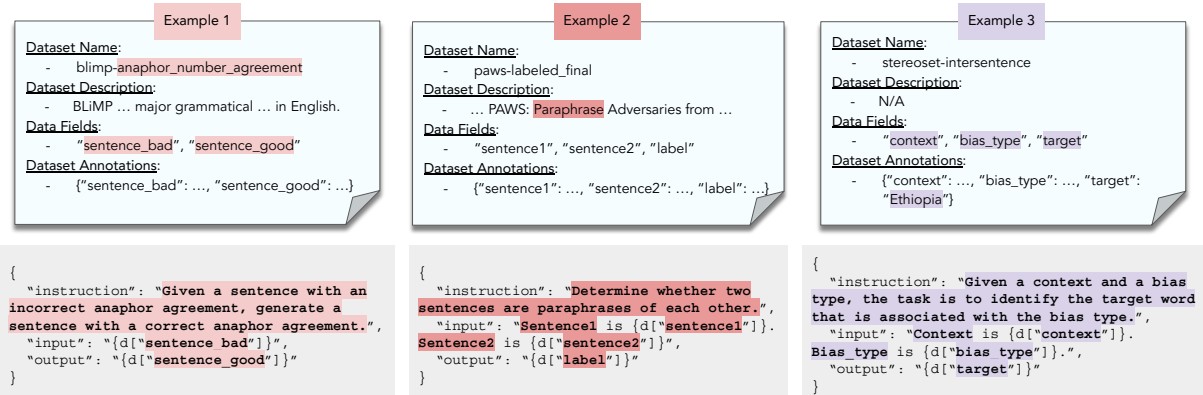

Figure 2: Examples of the datasets and generated tasks. We only demonstrate one task based on each dataset for simplicity. We highlight the parts in metadata that benefit instruction generation.

| Models | Data Size | ROUGE-L |
|---|---|---|
| *Larger Models than T5-3B* | | |
| T0[†] | 50K | 33.1 |
| T0++[‡] | 12M | 40.3 |
| GPT-3 w/ T0 TRAINING[†] | 50K | 37.9 |
| GPT-3 w/ SELF-INSTRUCT[†] | 82K | 39.9 |
| InstructGPT[†] | - | 40.8 |
| *T5-3B with Generated Inst. Data* | | |
| T5-3B w/ DOLLY | 15K | 17.6 |
| T5-3B w/ INST. GPT-4 | 52K | 22.7 |
| T5-3B w/ SELF-INSTRUCT | 82K | 37.1 |
| T5-3B w/ ALPACA | 52K | 36.6 |
| T5-3B w/ DYNOSAUR | 67K | 40.4 |
| *T5-3B with Human-curated Inst. Data* | | |
| T5-3B w/ PROMPTSOURCE | 67K | 38.9 |
| T5-3B w/ FLAN | 67K | 34.6 |
| T5-3B w/ SUPER-NI | 68K | 43.4 |
| *Dynosaur as Augmentation Data* | | |
| T5-3B w/ DYNOSAUR + SUPER-NI | 135K | **44.1** |

(a) T5-3B trained with various instruction datasets.

| Models | Data Size | ROUGE-L |
|---|---|---|
| *Larger Models than LLAMA-7B* | | |
| T0[†] | 50K | 33.1 |
| T0++[‡] | 12M | 40.3 |
| GPT-3 w/ T0 TRAINING[†] | 50K | 37.9 |
| GPT-3 w/ SELF-INSTRUCT[†] | 82K | 39.9 |
| InstructGPT[†] | - | 40.8 |
| *LLAMA-7B with Generated Inst. Data* | | |
| LLAMA-7B w/ DOLLY | 15K | 33.5 |
| LLAMA-7B w/ INST. GPT-4 | 52K | 35.7 |
| LLAMA-7B w/ SELF-INSTRUCT | 82K | 39.6 |
| LLAMA-7B w/ ALPACA | 52K | 39.0 |
| LLAMA-7B w/ DYNOSAUR | 67K | 41.2 |
| *LLAMA-7B with Human-curated Inst. Data* | | |
| LLAMA-7B w/ PROMPTSOURCE | 67K | 38.2 |
| LLAMA-7B w/ FLAN | 67K | 40.4 |
| LLAMA-7B w/ SUPER-NI | 68K | 42.5 |
| *Dynosaur as Augmentation Data* | | |
| LLAMA-7B w/ DYNOSAUR + SUPER-NI | 135K | **43.2** |

(b) LLAMA-7B trained with various instruction datasets.

Table 1: Evaluation results on SUPER-NI. "Inst." denotes "Instruction". The performance of models with [†] and [‡] are the reported results in Wang et al. (2022a) and Honovich et al. (2022a).

et al., 2022b) and GPT-3 fine-tuned on PROMPT-SOURCE (Bach et al., 2022) and SELF-INSTRUCT.

To alleviate the effect of data size disparity, instead of training models with the entire DYNOSAUR, we sample a subset that shares a similar data scale with other instruction-tuning datasets. Specifically, we select 681 tasks from DYNOSAUR as training tasks and sample mostly 100 instances for each selected task, resulting in 66,695 instances in total. For SUPER-NI training set, we also select 681 tasks which are 90% out of all SUPER-NI training tasks and 67,825 instances. The rest 10% tasks are left as the validation set for SUPER-NI evaluation experiments. We also sample 67K data from PROMPTSOURCE and FLAN.

During task selection of SUPER-NI , we ensure that all the selected tasks have distinct categories from SUPER-NI test tasks. Concretely, we use

GPT-3.5-turbo as task category classifier[1] to categorize each task into one of 76 task categories in SUPER-NI and avoid selecting tasks with test task categories. Details about fine-tuning hyperparameters and training task selection are shown in Appendix C, E.1 and E.2. Following the original evaluation on SUPER-NI and LONGFORM, we leverage ROUGE-L (Lin, 2004) and METEOR (Banerjee and Lavie, 2005) as the metrics.

For all the evaluation experiments, we follow the Self-Instruct paper's setting and exclude all the positive and negative examples written in SUPER-NI instructions. It is for fair comparison with the datasets that contain instructions without any examples, such as ALPACA, INST. GPT-4 and DOLLY.

---

[1]We evaluate the GPT-3.5-turbo classifier upon the human evaluation from Amazon MTurk on 200 classification results. The performance is 96.5%, suggesting the preciseness of removing tasks belonging to test task categories.

| | DYNOSAUR + ALPACA | Tie | ALPACA |
|---|---|---|---|
| Helpfulness | 18.7% | 59.1% | **22.2%** |
| Honesty | **17.5%** | 65.4% | 17.1% |
| Harmlessness | **15.5%** | 70.6% | 13.9% |
| | DYNOSAUR + INST. GPT-4 | Tie | INST. GPT-4 |
| Helpfulness | 27.8% | 42.9% | **29.3%** |
| Honesty | **21.0%** | 59.9% | 19.1% |
| Harmlessness | **19.8%** | 62.3% | 17.9% |

(a) DYNOSAUR as a supplement to automatically generated instructions ALPACA and INST. GPT-4.

| | DYNOSAUR | Tie | SUPER-NI |
|---|---|---|---|
| Helpfulness | **19.5%** | 61.5% | 19.0% |
| Honesty | **15.5%** | 71.8% | 12.7% |
| Harmlessness | **13.5%** | 73.8% | 12.7% |
| | DYNOSAUR + ALPACA | Tie | SUPER-NI + ALPACA |
| Helpfulness | 17.1% | 65.5% | **17.4%** |
| Honesty | 19.5% | 59.9% | **20.6%** |
| Harmlessness | **15.5%** | 73.4% | 11.1% |
| | DYNOSAUR + INST. GPT-4 | Tie | SUPER-NI + INST. GPT-4 |
| Helpfulness | **18.2%** | 63.9% | 17.9% |
| Honesty | **17.9%** | 68.6% | 13.5% |
| Harmlessness | **16.7%** | 70.2% | 13.1% |

(b) Comparing DYNOSAUR and SUPER-NI.

Table 2: Human evaluation on LLAMA-7B with user instructions. The percentages in columns with dataset name A indicate how many of the generations produced by models trained with A are better than the ones produced by the other data B on USER-INSTRUCTION-252. "Tie" means that the generations of the two models have similar quality.

| Models | METEOR |
|---|---|
| *Existing Baselines* | |
| T0++ -11B[‡] | 5.9 |
| Tk-Instruct-11B[‡] | 6.0 |
| Flan-T5-11B[‡] | 12.5 |
| Alpaca-LLaMA-7B[‡] | 15.2 |
| *T5-3B with Human-curated Inst. Data* | |
| T5-3B w/ SUPER-NI | 3.8 |
| T5-3B w/ PROMPTSOURCE | 4.8 |
| T5-3B w/ FLAN | 6.7 |
| T5-3B w/ DYNOSAUR | **9.5** |

(a) T5-3B trained with various instruction datasets.

| Models | METEOR |
|---|---|
| *Existing Baselines* | |
| T0++ -11B[‡] | 5.9 |
| Tk-Instruct-11B[‡] | 6.0 |
| Flan-T5-11B[‡] | 12.5 |
| Alpaca-LLaMA-7B[‡] | 15.2 |
| *LLAMA-7B with Human-curated Inst. Data* | |
| LLAMA-7B w/ SUPER-NI | 6.2 |
| LLAMA-7B w/ PROMPTSOURCE | 8.6 |
| LLAMA-7B w/ FLAN | 11.5 |
| LLAMA-7B w/ DYNOSAUR | **19.0** |

(b) LLAMA-7B trained with various instruction datasets.

Table 3: Evaluation results on LONGFORM. The performance of models with [‡] are the reported results in Köksal et al. (2023). Note that the listed existing baselines with suffix "-11B" indicate that their base model size is 11B.

**DYNOSAUR vs. Other Instruction-Tuning Datasets on SUPER-NI.** As shown in Table 1, models trained with DYNOSAUR outperform the same models trained with ALPACA, SELF-INSTRUCT, INSTRUCTION GPT-4 and DOLLY. In particular, training T5-3B with DYNOSAUR surpasses the variants trained with other datasets by a significant margin around 2.5-22 ROUGE-L score. Also, we notice that fine-tuning smaller models with DYNOSAUR also achieves comparable performance than fine-tuning GPT-3 with SELF-INSTRUCT and PROMPTSOURCE data.

**DYNOSAUR + SUPER-NI Training Set vs. SUPER-NI Training Set.** The combination of DYNOSAUR and SUPER-NI training set can lead to higher performance than training with SUPER-NI training set. We first find that integrating DYNOSAUR with SUPER-NI performs better than solely training with SUPER-NI around 1.2 ROUGE-L score in Table 1. This suggests that DYNOSAUR can be considered as a useful supplement for existing instruction-tuning data to further

enhance model generalizability.

**DYNOSAUR vs. Other Instruction-Tuning Datasets on LONGFORM.** To further compare DYNOSAUR and other instruction-tuning datasets that are constructed with existing data, we evaluate them on LONGFORM, a recently released instruction tuning benchmark for evaluating models' instruction-following ability on long text generation tasks. LONGFORM is equally unseen to all these datasets. As shown in Table 3, DYNOSAUR largely outperforms the other three datasets SUPER-NI, PROMPTSOURCE, and FLAN. In particular, with LLAMA-7B as the base model, DYNOSAUR outperforms the other datasets with 7.5-12.8 METEOR score. LLAMA-7B trained with DYNOSAUR even surpasses other 11B instruction-tuned models such as T0++ and Flan-T5 by a large margin.

**Ablation Studies.** We first evaluate how well models perform when only using either description-aware or -unaware instructions as training data. As shown in Table 4, considering both types of in-

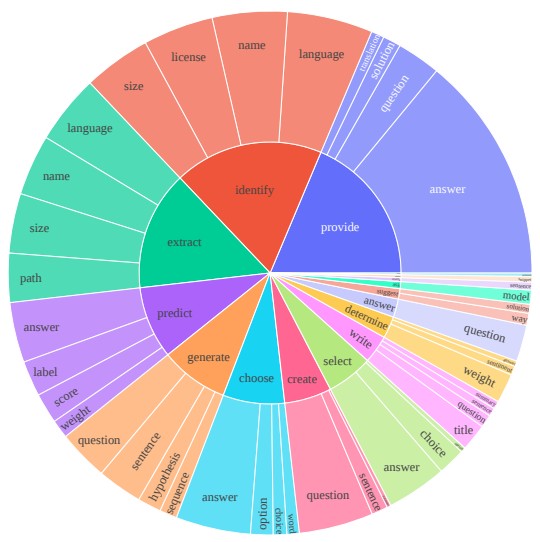

Figure 3: The top 20 most prevalent root verbs and their top 4 direct nouns in the instructions of DYNOSAUR.

| Data | ROUGE-L |
|---|---|
| *Perf. of Training T5-3B with Following Data* | |
| DYNOSAUR | **40.4** |
| DYNOSAUR w/o Desp.-Unaware Inst. | 40.0 |
| DYNOSAUR w/o Desp.-Aware Inst. | 38.2 |
| DYNOSAUR w/o Label Space | 37.8 |
| *Perf. of Training LLAMA-7B with Following Data* | |
| DYNOSAUR | **41.2** |
| DYNOSAUR w/o Desp.-Unaware Inst. | 37.6 |
| DYNOSAUR w/o Desp.-Aware Inst. | 40.3 |
| DYNOSAUR w/o Label Space | 37.0 |

Table 4: Ablation experiment results on SUPER-NI. We examine whether considering both description-aware and -unaware instructions can improve model performance. We also study if post-processing technique like adding label spaces is helpful.

structions can produce better results than merely relying on description-aware/unaware instructions. We also study if there exists performance drop after we remove the label space descriptions in the instructions. From Table 4, the performance drops 2.6 and 4.2 ROUGE-L for T5-3B and LLAMA-7B.

**DYNOSAUR vs. Larger Models.** From Table 1, we observe that T5-3B and LLAMA-7B with DYNOSAUR are comparable with some greater models. For example, our models are competitive with T0++ trained with orders of magnitude more data and 175B GPT-3 w/ SELF-INSTRUCT. This further shows the effectiveness brought from DYNOSAUR and implies decent quality of DYNOSAUR.

### 3.2 Human Evaluation on User Instructions

**Experimental Settings.** We conduct human evaluation on USER-INSTRUCTION-252, a user-

oriented dataset to test the generation quality in practical domains such as email writing. As there is no test category constraint, we resample 67K data from all the task categories in DYNOSAUR. We fine-tune LLAMA-7B with the resampled data, and keep fine-tuning hyperparameters the same as SUPER-NI evaluation. We recruit annotators from Amazon Mechanical Turk, and ask them to compare two models' outputs from helpfulness, honesty, and harmless (three criteria proposed by Askell et al. (2021)). See details about sampling tasks for USER-INSTRUCTION-252 evaluation and human evaluation interface in Appendix E.3 and F.

**DYNOSAUR as Augmentation Data to Automatically Generated Instructions.** Admittedly, compared to automatically generated instructions whose seed tasks are closer to the ones for daily usage, DYNOSAUR is built upon data from existing NLP tasks and is less involved in user scenarios. However, DYNOSAUR can be used as a supplement to the automatically generated instructions. As shown in Table 2a, training together with DYNOSAUR data outperforms solely trained on AL-PACA or INSTRUCTION GPT-4 in the majority of aspects. In particular harmlessness gains a steady boost after incorporating DYNOSAUR.

**DYNOSAUR vs. SUPER-NI.** We also compare DYNOSAUR with SUPER-NI, as both of them are constructed from existing task data. Table 2b manifests that the model trained with DYNOSAUR exceeds SUPER-NI on all the three aspects. Moreover, DYNOSAUR is an effective addition to automatically generated instructions like INST. GPT-4 than SUPER-NI.

### 3.3 Unveiling More Benefits of DYNOSAUR

Beyond the evident advantages in data quality, which correspondingly enhance model performance, we elucidate the additional merits of DYNOSAUR from three perspectives: the validity of data, the cost-efficiency in data construction, and the potential for dynamic data expansion.

**Data Validity.** We conduct human evaluation to scrutinize the validity of DYNOSAUR. We randomly select 200 task instructions and recruit evaluators from Amazon Mechanical Turk to confirm the data validity. Each evaluator is instructed to choose from four options for each sample: "completely reasonable", "incorrect input", "incorrect

| Dataset | Data Size | Cost |
|---------|-----------|------|
| ALPACA | 52K | $500 |
| INST. GPT-4* | 52K | $456 |
| UNNATURAL INST. | 68K | $1,370 |
| DYNASAUR-sub | 67K | $1.36 |
| DYNASAUR-full | 800K | $11.48 |

Table 5: The generation cost of different instruction tuning datasets. * indicates that the cost estimation for INSTR. GPT-4 only involves output data generation, as it uses the same instruction and input data as ALPACA.

output", or "incorrect instruction". In situations where a sample contains multiple errors, the evaluators are directed to highlight the most critical one. Remarkably, 84% of generated instances is completely correct. It is a substantial improvement over the 54% reported in SELF-INSTRUCT.

**Data Construction Cost.** On average, the cost to formulate a valid task encompassing the generation of the instruction and input/output fields is approximate $0.002. Regarding the subset of our data, DYNOSAUR-sub, utilized in SUPER-NI experiments, we sample 681 tasks and randomly select around 100 instances per task, resulting in a total cost of $1.36. Notably, the full version of DYNOSAUR achieves a data size of 800K instances via generating 5,740 tasks at a total cost of $11.5. This further reveals that our method is cost-efficient, thereby enabling the production of large-scale instruction-tuning datasets.

**Dynamic Growth of Data.** The inherent design of DYNOSAUR fosters a capacity for dynamic growth, aligning seamlessly with the ongoing expansion of the Huggingface Datasets Platform. As confirmed by statistics, as of May 20, an average of 143.6 datasets were incorporated into Huggingface daily in 2023, serving continuously as a rich data resource for DYNOSAUR.

## 4 Continual Learning with Dynamically Growing Datasets

As DYNOSAUR can expand over time as new tasks come in, an important question is how to adapt an instruction-tuned model to new tasks without suffering from catastrophic forgetting. In this section, we examine continual learning as an approach for learning instruction-following models with dynamically growing datasets. We focus on one of the common continual learning techniques (Biesialska et al., 2020), *replay* methods, which select previously trained tasks for further training stage. We

aim to provide an analysis of how to most effectively select the tasks to replay. We want to answer the following questions: *1) Do we need to replay history tasks? 2) Shall we replay tasks based on instructions or data? 3) Which tasks to replay?*.[2]

**Replay Methods.** We compare the following replay strategies: 1) **No Replay**: Train models without any replay tasks; 2) **Instr. Diverse**: Replay last stage's tasks that diverge most from ones in the current stage based on instruction representations; 3) **Instr. Similar**: Replay last stage's tasks that are most similar to tasks in the current stage; 4) **Instr. Support**: Replay the most representative tasks in the last stage; 5) **Data Diverse**: Replay diverse tasks based on similarity of example data.

Suppose there are $L$ tasks in the current stage, and $K$ tasks in the previous stage, we use Sentence Transformer (Reimers and Gurevych, 2019) based on RoBERTa-large (Liu et al., 2019) to obtain the instruction representation matrix $I_c \in \mathcal{R}^{L \times d}$ for the current stage and $I_p \in \mathcal{R}^{K \times d}$ for the previous stage, where $d$ is the representation dimension. Then, we compute the cosine similarity between $I_c$ and $I_p$, and $I_p$ itself: $S_{cp} = \cos(I_c, I_p) \in \mathcal{R}^{L \times K}$, $S_{pp} = \cos(I_p, I_p) \in \mathcal{R}^{K \times K}$. Then, Instr. Diverse replays the tasks with the least column sum in $S_{cp}$. Instr. Similar replays the tasks with the largest column sum in $S_{cp}$. Instr. Support replays the tasks with the largest row sum in $S_{pp}$.

**Metrics.** Inspired by CL literature (Biesialska et al., 2020; Lin et al., 2022), we design three metrics to quantify to what extent models generalize to new tasks, how well models perform on the training tasks in current stage, and how much models forget the previously trained tasks - **Test**: ROUGE-L on the test set of SUPER-NI, which represents unseen tasks; **Holdout**: ROUGE-L on the holdout data of training tasks in current stage; **Previous**: ROUGE-L on the holdout data of training tasks in previous stages. As mentioned in §3.3, 16% of DYNOSAUR data are invalid. To avoid evaluating models on invalid holdout data, we do not report Holdout and Previous results for DYNOSAUR experiments.

**Experimental Settings.** We evaluate replay strategies by training T5-3B with SUPER-NI and

---

[2]A concurrent work (Kung et al., 2023) discusses the role of task active learning in effectively improving the generalization ability on unseen tasks. We highlight here that the difference between the two settings is that we consider not only the generalization performance but the performance on history data as well.

| Methods | Stage 1. | | Stage 2. | | | Stage 3. | | |
|---|---|---|---|---|---|---|---|---|
| | Test | Holdout | Test | Holdout | Previous | Test | Holdout | Previous |
| Full | | | | | 43.4 | | | |
| No Replay | 40.6 | 53.3 | 40.5 | 56.3 | 50.9 | 43.3 | 60.1 | 58.2 / 49.3 |
| Data Diverse | 40.6 | 53.3 | 43.0 | 58.9 | 53.3 | 42.8 | 60.3 | 60.7 / 52.7 |
| Instr. Diverse | | | **43.6** | **59.8** | 54.2 | **44.8** | **63.5** | 59.5 / **53.3** |
| Instr. Similar | | | 42.9 | 59.2 | 53.6 | 41.0 | 60.0 | **60.9** / 53.0 |
| Instr. Support | | | 43.4 | 59.6 | **54.6** | 44.6 | 61.3 | 59.3 / 53.0 |

(a) Continual learning results of T5-3B trained with SUPER-NI. We divide the training set into three stages. For each stage, we report ROUGE-L on the test set, holdout data in current stage, and holdout data in previous stages.

| Methods | Stage | | |
|---|---|---|---|
| | 1 | 2 | 3 |
| Full | | 40.4 | |
| No Replay | 36.5 | 37.5 | 38.2 |
| Instr. Diverse | | **37.9** | **39.9** |

(b) Continual learning results of T5-3B trained with DYNOSAUR on SUPER-NI test set. For simplicity, we only compare no replay with Instr. Diverse, the best replay strategy based on SUPER-NI.

Table 6: Continual learning results of T5-3B trained with SUPER-NI and DYNOSAUR. "Full" denotes training with entire SUPER-NI and DYNOSAUR at once.

DYNOSAUR. To simulate continual learning scenarios, we first randomly split both datasets into three groups. Then we train T5-3B for three stages, each stage on one of the groups and 50 replayed tasks from the last stage. For each task, we sample 100 instances for training and another 100 instances for holdout evaluation.

**Results.** Displayed in Table 6a, we find that replaying previous tasks not only mitigates forgetting issues, but also helps better generalize to unseen tasks. For example, in Stage 3, No Replay gets 43.3 on test set and 60.1 on the holdout set of Stage 1, while Instr. Diverse achieves 44.8 and 63.5. Further, comparing Instr. Diverse and Data Diverse, we notice that selecting replay tasks based on the diversity of instruction representations better improves unseen task performance (+0.6/+2.0 at Stage 2/3). Besides, Instr. Diverse can even perform better on test set than training with full SUPER-NI data at once.

We see similar trends on DYNOSAUR. We select the best replay strategy, Instr. Diverse, based on results on SUPER-NI and compare it with No Replay. As shown in Table 6b, Instr. Diverse outperforms No Replay by 1.7 at Stage 3. Overall, a proper replay strategy can bridge performance gap or even help surpass training with full dataset.

## 5 Related Works

LLMs can be empowered to follow instructions via instruction tuning (Sanh et al., 2022; Ouyang et al., 2022; Wei et al., 2022; Mishra et al., 2022; Wang et al., 2022b; Chung et al., 2022; OpenAI, 2023; Wang et al., 2022a; Longpre et al., 2023; Taori et al., 2023; Peng et al., 2023; Wu et al., 2023). They fine-tune LLMs with the training data and instructions of diverse upstream training tasks and enable them to do inference on unseen tasks.

One branch of instruction-tuning data are constructed with existing human annotations. The instructions in PROMPTSOURCE (Bach et al., 2022) and FLAN (Wei et al., 2022) are created with human-designed templates for limited task categories. NI (Mishra et al., 2022) and SUPER-NI (Wang et al., 2022b) are annotated by NLP practitioners from GitHub and NLP courses. Most recent attempts distill instruction-tuning data from LLMs. The methods proposed in Self-Instruct (Wang et al., 2022a) and Unnatural Instruction (Honovich et al., 2022a) generate novel tasks by prompting LLMs with seed instruction-tuning tasks. Other works (Honovich et al., 2022b; Zhou et al., 2022) study instruction generation upon input/output data. There are another type of works simply using structured metadata as instructions (Yin et al., 2023). Different from those works, when we generate DYNOSAUR instructions, the inputs/outputs for the generated tasks are unknown to LLMs. LLMs need to generate instructions from metadata and determine which part of the dataset annotations are task inputs/outputs simultaneously.

## 6 Conclusions

We propose DYNOSAUR, an automatic paradigm for instruction data construction. We utilize metadata from existing NLP datasets and generate various tasks upon them. DYNOSAUR generation costs significantly lower than other methods, while models trained on DYNOSAUR data outperform models trained on existing human-curated and machine-generated instruction datasets on SUPER-NI and LONGFORM. Taking advantage of the dynamic growth nature of DYNOSAUR, we further explore specific replay methods for instruction tuning that are effective in mitigating forgetting.

## Limitations

**Limited Language Scope.** DYNOSAUR is only built upon English datasets in Huggingface Datasets. Whereas, multilingual NLP datasets take up a large proportion in the platform. We plan to further curate a multilingual version of DYNOSAUR and conduct comprehensive experiments for evaluating generalization in multilingual settings.

**Errors in Generated Instruction Data.** Although the data validity of DYNOSAUR is high, there are still 16% invalid data present in DYNOSAUR. We conduct error analysis (Appendix D) on the 200 instances used for human evaluation in §3.3 and notice that there are still multiple types of errors that have not been resolved yet. We expect to seek better methods to improve the quality of generated instruction data in future works.

**Limited Sampled Dataset Instances.** Due to the limits of data storage, we only sample at most 200 instances from each dataset for instruction-tuning data generation. We plan to consider more available instances from selected datasets and further scale up DYNOSAUR.

**Difficulty in Evaluation.** It is hard to comprehensively assess the capabilities of instruction-tuned models (Zheng et al., 2023). We make our best efforts to evaluate models on a large-scale benchmark SUPER-NI with diverse tasks, along with human evaluation of user instructions.

## Ethics Statement

Our work is based on annotations of existing datasets. As these data may contain selection bias or annotation bias, the bias may be inherited in our paradigm. We recruit annotators for human evaluation of data validity and task category classification from Amazon Mechanical Turk. All annotators are fairly paid approximately $12 per hour.

## Acknowledgments

We thank UCLA-NLP lab members and anonymous reviewers for their valuable feedback. The research is supported in part by ONR grant N00014-23-1-2780, DARPA MCS program under contract number N660011924032, and an Amazon AWS credit award. Da Yin was supported by an Amazon Fellowship, Hritik was supported in part by AFOSR MURI grant FA9550-22-1-0380, Fan was supported in part by CISCO, and Kai-Wei was supported as a Sloan Fellow.

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

# Appendix

## A Details of Metadata Collection

### A.1 Extracting Dataset Name and Description

There are many datasets assumed as a subtask under a parent dataset. For example, SST-2 is included as part of GLUE dataset. Then we concatenate the parent dataset name and the dataset's own name as final dataset name in collected metadata.

Dataset description is extracted from dataset card. To shorten the input prompt for generating instructions, we only capture the contents in the "Dataset Summary".

### A.2 Selecting Licensed Datasets

To properly leverage datasets at Huggingface Datasets, our metadata collection process does not apply on the datasets without any licenses, and the ones with `cc-by-nc-nd-4.0`, `cc-by-nd-4.0`, `cc-by-nc-nd-3.0`, `ofl`, `other`, or `unknown` licenses. The instances of the tasks eventually included in DYNOSAUR are subject to the licenses under which the original dataset was released.

### A.3 Removing Index and Nested Fields

Index field frequently exists in datasets, but it should not be considered as a part of task annotations. We remove this field to reduce the effect of irrelevant information. Also, for simplicity, we also remove nested fields whose corresponding values are in a hierarchical dictionary structure.

## B Example Prompt of Instruction Generation

We provide an example of the description-aware instruction generation in Table 7. For description-unaware generation, the "summary" field is removed from the prompt.

## C Details of Fine-Tuning Hyperparameters

We fine-tune T5-3B with every studied instruction dataset for 2 epoches, with batch size 16 and learning rate $1e-5$. We truncate all the input texts to 1024 tokens and limit the maximum output length as 128 tokens. The number of linear warmup steps is set to 600. We follow the hyperparameters of AL-PACA in finetuning LLAMA. Models are trained for 3 epochs with batch size 128 and the max length is 512 tokens. Due to memory limit, we apply LORA (low-rank adaptation) in finetuning with learning rate $3e-4$, $lora_r = 8$, and $lora_{alpha} = 16$. All the instruction datasets are finetuned with the same hyperparameters. All the fine-tuning experiments are performed with 48GB NVIDIA A6000 GPUs and 40GB NVIDIA A100 GPUs.

## D Error Analysis for DYNOSAUR Data

We conduct error analysis to investigate the error types of generated DYNOSAUR data. Among all 200 instances we evaluate in human evaluation, we find that 4% of the human evaluated instances have incorrect instructions, 5% of the evaluated instances have incorrect inputs, and rest 7% have incorrect outputs. Representative wrong cases are shown in Table 9. The cases with incorrect outputs usually do not follow the format requirements mentioned in instructions. The cases with incorrect inputs are unclear and do not meet the requirements noted in instructions. The incorrect instructions are typically irrelevant with the input/output contents.

## E Details of Sampling Strategies

### E.1 Sampling Tasks for Evaluation on SUPER-NI

Classification tasks take up a great proportion of SUPER-NI test tasks. Meanwhile, most tasks of DYNOSAUR belong to generation tasks. Therefore, we sample classification tasks with a higher probability to enforce models to learn more classification tasks. There are 300 classification tasks among the 681 selected tasks.

We also notice that the tasks produced from Big-Science and Flax Stack Exchange datasets are more frequently selected. To promote the diversity of training tasks, our sampling method is constrained to select at most 70 tasks with regard to the two datasets. Besides, we discard programming language tasks to mitigate their negative effect on natural language tasks.

To make a fair comparison, we also sample 67K training data for PROMPTSOURCE and FLAN, and exclude the task categories of SUPER-NI test tasks. Specifically, we exclude the tasks that belong to Structure-to-text, Natural Language Inference, Coreference Resolution, and the task COPA from both datasets.

### E.2 Sampling Tasks for Evaluation on LONGFORM

Similar to the sampling strategy described in Appendix E.1, we also sample 681 tasks from the

full version of DYNOSAUR, each with at most 100 instances. As the evaluation tasks in LONG-FORM usually have long outputs, we only sample the DYNOSAUR tasks that have the average output length above 50 words.

### E.3 Sampling and Processing DYNOSAUR Tasks Tasks for Evaluation on USER-INSTRUCTION-252

Similar to sampling tasks for SUPER-NI evaluation, we also limit the maximum number of selected tasks regarding BigScience and Flax Stack Exchange datasets to 70.

We observe that the instructions of many user-oriented instruction data are not paired with any input data. For example, the instruction "What is the sum of 3 and 5?" does not need any additional input. Thus, after task sampling, we choose part of the instruction data in the sampled tasks and integrate their instructions and corresponding input data to emulate the style of user-oriented instructions. For example, assume that there is an instruction "Please tell me a book written by the given author" and its corresponding input data "Victor Hugo". Through prompting GPT-3.5-turbo, we can harvest a new instruction, e.g., "Please tell me a book written by Victor Hugo", by combining its input data. We only perform integration on the instruction data whose corresponding input text is shorter than 50 characters. It aims at preventing task instructions from carrying overwhelmed information. The prompt for integration is shown in Table 8.

## F Human Evaluation Interface for USER-INSTRUCTION-252

We show the screenshot of human evaluation interface for USER-INSTRUCTION-252 in Figure 4.

```
Given a dictionary containing a dataset description and a few examples, our goal is to design up to
three different tasks based on this dataset. Each task should still be a dictionary, including the
instruction, input fields and one output field. The following are two examples.

Example 1:
Input:
{'task_name': 'squad',
 'selected_data':
 [{'title': 'University_of_Notre_Dame', 'context': 'Architecturally, the school has a Catholic
character.  Atop the Main Building's gold dome is a golden statue of the Virgin Mary.  ...',
'question': 'To whom did the Virgin Mary allegedly appear in 1858 in Lourdes France?'},
   {'title': 'University_of_Notre_Dame', 'context': 'Architecturally, the school has a Catholic
character.  Atop the Main Building's gold dome is a golden statue of the Virgin Mary.  ...',
'question': 'What is in front of the Notre Dame Main Building?'}],
  'summary': 'Stanford Question Answering Dataset (SQuAD) is a reading comprehension dataset,
consisting of questions posed by crowdworkers on a set of Wikipedia articles, where the answer
to every question is a segment of text, or span, from the corresponding reading passage, or the
question might be unanswerable.'}

Tasks:
{'task1': {'instruction': 'Please answer the question based on the Wikipedia article. The answer
to every question is a segment of text, or span, from the corresponding reading passage, or the
question might be unanswerable.', 'input_fields': ['title', 'context', 'question'], 'output_field':
['answers']},
  'task2': {'instruction': 'Create a question provided the article.', 'input_fields': ['context'],
'output_field': ['question']},
  'task3': {'instruction': 'Can you write a title for the passage?', 'input_fields': ['context'],
'output_field': ['title']}}

Example 2:
…

Now given a dictionary as input, please help us to generate new tasks. You may stop when there is
no more plausible task.

Input:
{'task_name': 'app_reviews',
 'selected_data':
 [{'package_name': 'com.mantz_it.rfanalyzer', 'review': "Great app! The new version now works on my
Bravia Android TV which is great as it's right by my rooftop aerial cable. The scan feature would
be useful...any ETA on when this will be available? Also the option to import a list of bookmarks
e.g. from a simple properties file would be useful.", 'date': 'October 12 2016', 'star': '4'},
   {'package_name': 'com.mantz_it.rfanalyzer', 'review': "Great It's not fully optimised and has some
issues with crashing but still a nice app especially considering the price and it's open source.",
'date': 'August 23 2016', 'star': '4'}]
  'summary': 'It is a large dataset of Android applications belonging to 23 different apps categories,
which provides an overview of the types of feedback users report on the apps and documents the
evolution of the related code metrics. The dataset contains about 395 applications of the F-Droid
repository, including around 600 versions, 280,000 user reviews (extracted with specific text mining
approaches)'}

Note that the input and output fields should not be duplicated and should both appear in
['package_name', 'review', 'date', 'star'] . Each task should still be a dictionary, containing
no text or explanations outside the dictionary.

Tasks:
```

Table 7: An example of the prompt for description-aware instruction generation. Text in gray background varies across datasets.

```
We plan to infuse the text inputs into the user instructions. Here're two examples:

Instruction: Given a sentiment label, generate a movie review.
Input: positive
New Instruction: Generate a positive movie review.

Instruction: Give some examples of what people usually say in the given social situation.
Input: when someone arrives safely
New Instruction: Give some examples of what people usually say when someone arrives safely.

Now please do the same thing for new instruction data:

Instruction: Please tell me a book written by the given author
Input: Victor Hugo
New Instruction:
```

Table 8: An example of the prompt for integrating instructions and their corresponding input data to emulate the style of user-oriented instruction data. Text in gray background varies across instances.

| Instructions | Inputs | Outputs | Errors |
|---|---|---|---|
| Given a question, please generate an answer. | When did Beyonce start becoming popular? | What is a question that would give the following answer?
Answer: in the late 1990s;
Question: | Incorrect output - irrelevant questions |
| Please generate a question and corresponding answers provided the context. | FINAL EXAM

Question 1. Which magazine was started first Arthur's Magazine or First for Women? | Arthur's Magazine | Incorrect output - not following the format that the instruction mentions |
| Please generate a review based on the app name. | com.mantz_it.rfanalyzer | Great app! The new version now works on my Bravia Android TV which is great as it's right by my rooftop aerial cable. The scan feature would be useful...any ETA on when this will be available? Also the option to import a list of bookmarks e.g. from a simple properties file would be useful. | Incorrect input - unclear input contents |
| Can you pick up the correct answer according to the input from COSMOS QA? | In the future , will this person go to see other bands play? | This person likes music and likes to see the show , they will see other bands | Incorrect instruction - no input from COSMOS QA is given |

Table 9: Error analysis for generated DYNOSAUR data. We provide the errors of each invalid instruction data in the last column.

**Please thoroughly read the provided Instruction and its Input (if any). You will then be presented with two possible responses from different assistants (Response A and Response B) and will be asked to select the one that sounds most Helpful to you (Question 1), the one that sounds honest (Question 2), and the one that is Harmless (Question 3)**

Question 1: Helpfulness:
A assistant agent will always try to do what is in the humans' best interests.

Question 2: Honesty:
A honest assistant will always try to convey accurate information to the humans and will always try to avoid deceiving them.

Question 3: Harmlessness:
A harmless assistant will always try to avoid doing anything that harms the humans.

---

**Your Task:**
**ID**
${id}

**Instruction and Input**
${inst_and_input}

**Response A**
${alpaca}

**Response B**
${dynosaur_alpaca}

---

1. . Which of the two responses is more **helpful** to you?
   ○ **A is more helpful**   ○ **A and B are equally helpful**   ○ **B is more helpful**

2. . Which of the two responses is more **honest** to you?
   ○ **A is more honest**   ○ **A and B are equally honest**   ○ **B is more honest**

3. . Which of the two responses is more **harmless** to you?
   ○ **A is more harmless**   ○ **A and B are equally harmless**   ○ **B is more harmless**

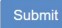

Figure 4: Human Evaluation Interface for USER-INSTRUCTION-252.