# OpenReview forum: "Dynosaur: A Dynamic Growth Paradigm for Instruction-Tuning Data Curation"
_EMNLP/2023/Conference — EMNLP 2023 Main_

### Official Review · Reviewer_eo7N · 2023-08-02

**Soundness:** 3

**Excitement:**

3: Ambivalent: It has merits (e.g., it reports state-of-the-art results, the idea is nice), but there are key weaknesses (e.g., it describes incremental work), and it can significantly benefit from another round of revision. However, I won't object to accepting it if my co-reviewers champion it.

**Paper Topic And Main Contributions:**

In this paper, the authors introduce a framework called Dynosaur which extends instruction tuned datasets using ChatGPT prompting as well as data and metadata compiled from the Huggingface Datasets Platform. This framework allows for much faster, cheaper and higher quality generation of large scale instruction-tuned datasets than other automatically created datasets like Alpaca and Self-Instruct. Using Dynosaur in combination with other instruction-tuning datasets such as SuperNI, Alpaca or Instruct-GPT4 leads to improvements. With data parity, improvements are still below carefully curated but less varied datasets like SuperNI.

**Questions For The Authors:**

- Do you have any insight into why harmlessness and honesty improves with DYNOSAUR? It seems quite surprising that its main gains are not in the “helpfulness” metric.
- Is there a difference in quality from the first task produced by ChatGPT to the third one in each example? Why did you predict 3 different tasks at once?


**Reasons To Accept:**

- Introduces an interesting setup for more accurate automatic instruction-tuning data generation.
- Presents an important evaluation of the generated data which finds high levels of correctness
- Thorough automatic and human evaluations over a range of instruction-tuning datasets.
- A small experiment concerning continual learning based on the fact that more datasets are continually added to the Huggingface platform.


**Reasons To Reject:**

There are no serious issues that I can see with this work but there are a few issues that could be grounds for rejection until more work is done on this paper:

- Writing needs more polishing
  - Minor grammar mistakes
    - Ungrammatical use of LLM throughout the paper (L074, L078, …)
    - Awkward sentence “while L new 133 training tasks are coming.” L133-134
    - “with T0++ trained with orders of magnitude data” L389
    - “implies descent quality of DYNOSAUR.” L392
    - “Moreover, DYNOSAUR is an effective addition to automatically generated instructions like INST. GPT-4 than SUPER-NI.” L426
    - Very confusing sentence
    - Figure 1 is very confusing and caption under-explains it
  - More serious writing mistakes
    - Too much detail in introduction, contributions should be clearer and more concise
    - L050, SuperNatural Instructions definitely uses previous annotated data, confusing sentence
    - Contentious statement given as fact: “an efficient fine tuning approach on par with normal fine-tuning.” L321


- Continual learning section seems rushed and takes space from very important error analysis of the data augmentation itself and more detailed dataset descriptions
  - Continual learning is a very rich research direction but the experiments presented here seem to lack contextualization (no baselines or discussion of previous work)
  - Could be removed from the paper or minimized greatly


**Reproducibility:**

4: Could mostly reproduce the results, but there may be some variation because of sample variance or minor variations in their interpretation of the protocol or method.

**Reviewer Confidence:**

3: Pretty sure, but there's a chance I missed something. Although I have a good feel for this area in general, I did not carefully check the paper's details, e.g., the math, experimental design, or novelty.

---

> ### Author Rebuttal · Authors · 2023-08-28
>
> Thanks for the valuable comments! We address the reviewer’s questions and concerns as follows:
>
> ### **Part 1: Polish writing:**
> We will address these editorial comments, proofread, and improve the writing of the paper in the revision.
>
> ### **Part 2: Discussion about continual learning (CL):**
> - The continual learning section aims to serve as an analysis (instead of proposing new methods) to show that our data curation process can be combined with CL techniques to deal with the ever-expanding new task data without hurting history tasks much. We hope the results will serve as a starting point and bring up an important research question.
> - We are aware that there are other techniques in CL besides replaying. However, replaying works on the data end which is perfectly coupled with our data curation technique. Also, it is worth noticing that the variants investigated are all valid baselines that are partially included in the literature cited by the paper.
> - We thank the reviewer for suggesting additional literature surveys in CL and error analysis in the data augmentation method. With the extra page upon acceptance, we will be able to better coordinate the CL section and include more literature context around CL.
>
> ### **Part 3: Questions for the authors:**
> **Do you have any insight into why harmlessness and honesty improves with Dynosaur?**
>
> Honesty is a dimension that requires models to generate faithful responses to the instructions instead of hallucinating. Since Dynosaur data is generated based on existing human annotations, Dynosaur data are likely to suffer less from the hallucination. Adding Dynosaur data may help mitigate the hallucination issue. For harmlessness, we find that annotators usually label repetitive outputs as harmful ones. Since the outputs of Dynosaur data are generally concise, it will not encourage the model to generate too long and repetitive responses. Therefore, adding Dynosaur may also help improve the harmlessness.
>
> **Is there any quality difference between the first generated task to the third task in each example? Why did you predict 3 different tasks at once?**
>
> We don’t notice significant difference in quality from the first task to the third one in each example. We choose to predict three tasks at once because the task difference can be better guaranteed by asking LLMs to generate three tasks at the same time. Based on our experience, it is hard to control the three tasks to be different if we generate the tasks one by one three times.

---

### Official Review · Reviewer_Dkdk · 2023-08-05

**Soundness:** 4

**Excitement:**

4: Strong: This paper deepens the understanding of some phenomenon or lowers the barriers to an existing research direction.

**Paper Topic And Main Contributions:**

The paper studies automatic data curation for instruction tuning. The authors design a system called DYNOSAUR that uses LLM to generate task instructions based on the metadata from existing NLP datasets.  DYNOSAUR has low construction costs, shows better instruction-tuning performance than other baseline datasets of comparable sizes, and supports adding new instruction data to improve models. The authors further adopt continual learning on top of the instruction-tuning datasets and show the diverse instruction embeddings help reduce forgetting and better generalization.

The main contribution of DYNOSAUR is its data curation advantages in three aspects: low conversion costs, the effectiveness of instruction-tuning data, and support for new instruction data.

**Reasons To Accept:**

The idea of using the metadata in existing large-scale NLP datasets is novel and effective. Experimental evaluation is comprehensive and strong.

**Reasons To Reject:**

It is good that the generation of the instruction tuning dataset is using metadata of existing NLP datasets. However, for domain-specific datasets that often have no such metadata, this kind of instruction-tuning data would be difficult to obtain. The technique described in the paper could fail.

**Reproducibility:**

4: Could mostly reproduce the results, but there may be some variation because of sample variance or minor variations in their interpretation of the protocol or method.

**Reviewer Confidence:**

3: Pretty sure, but there's a chance I missed something. Although I have a good feel for this area in general, I did not carefully check the paper's details, e.g., the math, experimental design, or novelty.

---

> ### Author Rebuttal · Authors · 2023-08-28
>
> Thanks for the comments and nice words!
>
> **We will first clarify the definition of metadata.** As shown in Section 2.1, dataset metadata contains 1) dataset name, 2) dataset description, and 3) data fields and dataset annotations. All the datasets we select have metadata since they all have their names, data fields, and annotations. However, some of them don’t have dataset descriptions on their Huggingface dataset page. The instruction-tuning data generation for those datasets without dataset description will be based on description-unaware generation (See Section 2.2). When there’s no dataset description, our method can still generate instruction-tuning data based on the other metadata information (e.g., dataset name, data fields, and annotations).
>
> Example 3 in Figure 2 demonstrates a case for the description-unaware generation. There is no dataset description for the dataset called “stereoset-intersentence”, but it has other metadata information like dataset name, data fields, and annotations. The method can leverage these information to generate a new task instruction “Given a context and a bias type, the task is to identify the target word that is associated with the bias type.” We will make the description of our methods clearer in the future version.

---

### Official Review · Reviewer_FuNK · 2023-08-12

**Soundness:** 4

**Excitement:**

4: Strong: This paper deepens the understanding of some phenomenon or lowers the barriers to an existing research direction.

**Paper Topic And Main Contributions:**

The paper "DYNOSAUR: A Dynamic Growth Paradigm for Instruction-Tuning Data Curation" proposes a method to automatically generate instruction-tuning datasets by combining LLMs with existing NLP datasets from the huggingface datasets hub. To achieve this, the authors propose to follow works like Promptsource and Super-NaturalInstructions (SNI) and construct prompt templates that transform each instance of a supervised NLP dataset into an instruction-tuning instance. As their main contribution, they replace the crowd sourcing step that Promptsource and SNI used by ChatGPT. For each NLP dataset from the huggingface datasets hub, they prompt ChatGPT with in-context learning to construct a prompt template based on the dataset's metadata. By applying this methodology, the authors obtain the proposed dataset, Dynosaur. They evaluate Dynosaur in an automated evaluation on SNI and in a human-judged evaluation on User-Instruction-252. On SNI the authors find, that T5-3B/Llama fine-tuned on Dynosaur outperforms T5-3B/Llama fine-tuned on other LLM-generated instruction tuning datasets, but performs worse than T5-3b fine-tuned on SNI.  Combining Dynosaur and SNI however, performs better than SNI alone. On User-Instruction-252, the authors find that when combining other instruction-tuning datasets with Dynosaur, the fine-tuned models are rated a bit higher in honesty and harmlessness but lower in helpfulness. Finally, the authors show that replay-based continual learning methods allow T5-3B to learn from new datasets added to Dynosaur without catastrophic forgetting.

**Questions For The Authors:**

- Will code/data be made available?
- Why is performance of T5-3b only 43.4 on SNI? If I read the SNI paper right, it reports 54.3 in Figure 3.
- L922-923: What are the tasks from BigScience and Flax Stack Exchange? Are there references for these efforts available that you could cite?
- Are the results from a single run? If so, can some of the observed differences in scores be explained through high standard deviations?
- How many instances were used for the human evaluation?

**Reasons To Accept:**

- The prospect of using LLMs to create template-based instruction-tuning datasets is exciting, because the creation of comparable datasets have been large-scale efforts. For instance, both SNI and Promptsource each have over 40 authors. Dynosaur, on the other hand, could be created by spending less than $12 for the ChatGPT API.
- Using the proposed technique for data creation could potentially unlock the simple generation of large-scale instruction tuning datasets for languages other than English and for specalized domains.
​

**Reasons To Reject:**

- I am not entirely convinced by the experimental results. That, when tested on SNI, models trained on Dynosaur oupterform those trained on Alpaca, Dolly, Self-instruct and Inst. GPT-4 is not particularly surprising. That is because Dynosaur's data distribution is arguably much more similar to SNI than that of the other datasets, which are more specialized for an interactive setting. A fairer comparison would have been to compare to SNI, T0 and/or Flan on a dataset that is equally "unseen" for all models.
- I have related concerns regarding the human evaluation. The observed differences are rather small and adding Dynosaur to Alpaca and Inst. GPT-4 even leads to a drop in judged helpfulness, which is arguably the most important metric for an interative assistant. As far as I could see, the authors did not report any metrics on the reliability of the human evaluation which makes it hard to judge whether the observed differences are due to random fluctuations.
- In summary, it's hard to judge from the experiments whether Dynosaur really can lead to consistent improvements over comparable instruction tuning datasets. In my opion, systematically showing that Dynosaur is on par with SNI and Promptsource would make the paper much stronger, because the cost savings in the creation process would be a decisive advantage alone.
​

**Reproducibility:**

3: Could reproduce the results with some difficulty. The settings of parameters are underspecified or subjectively determined; the training/evaluation data are not widely available.

**Reviewer Confidence:**

4: Quite sure. I tried to check the important points carefully. It's unlikely, though conceivable, that I missed something that should affect my ratings.

**Typos Grammar Style And Presentation Improvements:**

- L336: "67,825" -> "67,825 instances"
- L389: "orders of magnitude data" -> "orders of magnitude more data" [?]

---

> ### Author Rebuttal · Authors · 2023-08-28
>
> Thanks for the constructive comments and nice words for our cost-saving advantage! We address the comments raised by the reviewer below.
>
> ### **Part 1: A fairer comparison would have been to compare to SNI, T0 and/or Flan on a dataset that is equally "unseen" for all models:**
> Our claim is that by generating data-driven instruction-tuning datasets, the model learns better zero-shot ability. Therefore, we compare Dynosaur with the datasets like Alpaca and Inst. GPT-4, which are not generated based on metadata of existing corpus.
>
> However, we acknowledge the reviewer's concern. To demonstrate the effectiveness of our generated instruction-tuning datasets, we add experiments as follows:
> 1. We fine-tune T5-3B and LLAMA-7B with two additional instruction-tuning datasets PromptSource and Flan and test the fine-tuned models on SNI. **Results show that Dynosaur can help to achieve better performance, compared with PromptSource and Flan**.
> 2. We evaluate the models trained with Dynosaur, SNI, PromptSource and Flan on an instruction-tuning evaluation dataset Longform [1]. Longform is equally unseen to all the aforementioned datasets. **Dynosaur largely outperforms the other three datasets. In particular, with LLAMA-7B as the base model, Dynosaur outperforms the other datasets with 7.5-12.8 METEOR score**.
> 3. We evaluate the models trained with Dynosaur, SNI, PromptSource and Flan on User-Instruction-252. **Dynosaur consistently outperforms the others on almost all the dimensions**.
>
> For a fair comparison, we take the same numbers of training data, 67K, from the aforementioned instruction tuning datasets.
>
> [1] Köksal et al., 2023, LongForm: Optimizing Instruction Tuning for Long Text Generation with Corpus Extraction, Arxiv.
>
> —---------------------------------------
>
> Our results on SNI test set are shown as follows:
>
> |   **Metric: ROUGE-L**     | **T5-3B** | **LLAMA-7B** |
> |----------|-------|----------|
> | Dynosaur |   40.0 ± 0.2    |   40.4 ± 0.9      |
> | PromptSource       |   38.7 ± 0.2     |     37.4 ± 0.5     |
> | Flan     |   34.1 ± 0.3    |   39.8 ± 0.8       |
> | SNI      |   42.6 ± 0.8   |   42.5 ± 0.2     |
>
> We find that Dynosaur can bring better performance than PromptSource and Flan for both base models, T5-3B and LLAMA-7B. It shows that our Dynosaur dataset can achieve comparable performance with expensive crowdsourced instruction tuning data built with much expert effort. Although Dynosaur underperforms SNI-train a bit, SNI-train shares a very similar instruction text style with SNI-test, SNI-train is more like in-domain instruction-tuning data and thus brings better results on SNI-test. Later on, we show that SNI has worse transferability on other out-of-domain instruction tuning data. Also, note that SNI-train was annotated by 88 NLP practitioners from GitHub and NLP courses. The annotation cost would be significantly higher than Dynosaur.
>
> —---------------------------------------
>
> We also evaluate on Longform. Longform is a recently released instruction tuning benchmark for evaluating the model’s instruction-following ability on long text generation tasks. It is not evaluated in our original submission and we will include results on this dataset in the revision. The results are shown below:
>
> |  **Metric: METEOR**    | **T5-3B** | **LLAMA-7B** |
> |----------|-------|----------|
> | Dynosaur |   **9.5**    |    **19.0**      |
> | PromptSource       |  4.8    |    8.6      |
> | Flan     |   6.7    |    11.5      |
> | SNI      |   3.8   |   6.2     |
>
> Due to the time cost of generating long texts, for each of our studied data, we only evaluate on one random seed. We find that Dynosaur outperforms SNI, PromptSource and Flan for T5-3B. The performance gaps become greater when we use LLAMA-7B as our base model, which is 12.8, 7.5, and 10.4.
>
> —---------------------------------------
>
> We further conduct human evaluation on User-Instruction-252. The comparison results are shown below:
>
> |              | **Dynosaur**  | **Tie**   | **SNI**   |
> |--------------|-----------|-------|-------|
> | Helpfulness  | **19.5%** | 61.5% | 19.0% |
> | Honesty      | **15.5%** | 71.8% | 12.7% |
> | Harmlessness | **13.5%** | 73.8% | 12.7% |
>
> |              | **Dynosaur**  | **Tie**   | **PromptSource**   |
> |--------------|-----------|-------|-------|
> | Helpfulness  | **20.6%** | 63.9% | 15.5% |
> | Honesty      | **19.0%** | 65.5% | 15.5% |
> | Harmlessness | **19.0%** | 62.3% | 18.7% |
>
> |              | **Dynosaur**  | **Tie**   | **Flan**   |
> |--------------|-----------|-------|-------|
> | Helpfulness  | **21.0%** | 58.4% | 20.6% |
> | Honesty      | 16.3% | 71.8% | **16.7%** |
> | Harmlessness | **15.5%** | 71.0% | 13.5% |
>
> This dataset is also equally unseen to all of our studied data, Dynosaur, SNI, PromptSource and Flan. We find that Dynosaur consistently obtains better performance on almost all the dimensions than SNI, PromptSource and Flan.
>
> Overall, all of these experiments show that Dynosaur is on par with or even better than SNI, PromptSource and Flan on diverse instruction-tuning evaluation sets.
>
> ### **Part 2: Helpfulness evaluation for the human evaluation experiments**:
> We conduct a deeper analysis on the helpfulness evaluation. The possible reason why adding Dynosaur results in lower helpfulness scores is that models trained with Dynosaur and Alpaca/Inst. GPT-4 generate shorter responses, but people prefer longer texts as more helpful texts. This phenomenon is discussed in the AlpacaFarm paper [2]. We show a typical case observed in our experiment to further explain the phenomenon:
>
> Instruction: Enter the words that satisfy the given condition. Input: 5 Countries that Start with S.
>
> **Response of LLAMA-7B w/ Dynosaur + Alpaca**: 1. Sweden 2. Singapore 3. South Africa 4. Spain 5. Switzerland
>
> **Response of LLAMA-7B w/ Alpaca**: Here is a list of 5 countries that start with the letter "S": 1. Saudi Arabia 2. Senegal 3. Sierra Leone 4. Singapore 5. South Sudan
>
> Both models provide five countries that start with “S”. We verify the correctness of all the countries. The main difference between the two model outputs is length - the responses of LLAMA-7B w/ Alpaca are longer than the other because the response attaches another sentence “Here is a list …” that doesn’t provide any additional useful information. However, 3 annotators think that the response of LLAMA-7B w/ Alpaca is more helpful.
>
> For our experiments, we observe that adding Dynosaur as additional training data to Alpaca and Inst. GPT-4 will decrease the response length by 11.2 and 14.6 words on average, respectively. To mitigate the effect of the length confounder and show the actual effectiveness of our Dynosaur data, we perform a new round of helpfulness evaluation. Different from the evaluation we did in the original paper, for the new evaluation, annotators are asked to score the helpfulness of each individual model response of the two models, instead of doing the comparison between the outputs of two different models. Each response is annotated by three annotators. We find that annotators regard 87.3% of LLAMA-7B w/ Dynosaur + Alpaca responses as helpful, while only 84.5% of LLAMA-7B w/ Alpaca are helpful. Also, annotators treat 89.3% of LLAMA-7B w/ Dynosaur + Inst. GPT-4 responses as helpful, while only 87.3% of responses generated by LLAMA-7B w/ Inst. GPT-4 are helpful. Both comparison results show that adding Dynosaur can still help improve the helpfulness of model responses.
>
> We conduct annotator inter-agreement experiments for the human evaluation discussed in Table 2 of our submission. Annotators we recruit for all the human evaluation experiments must finish at least 5000 HITs with at least 98% approval rate in Amazon MTurk. Any two annotators agree on 83.7% of all the instances. Cohen’s Kappa value is 0.68, which signifies substantial agreement according to the standards (https://www.statology.org/wp-content/uploads/2021/02/kappa1.png). The statistics further support the reliability of human evaluation studies. We will add the inter-agreement information in the following version.
>
> [2] Dubois et al., 2023, AlpacaFarm: A Simulation Framework for Methods that Learn from Human Feedback, Arxiv.
>
> ### **Part 3: Detailed questions for authors**
> #### **Will code and data be released?**
> As we stated in the abstract, code/data will be released upon acceptance.
>
> #### **Why is the performance of T5-3b only 43.4 on SNI? It is 54.3 in the SNI paper.**
> For our experiments, we follow the Self-Instruct paper’s setting [3]. We exclude all the positive and negative examples written in the SNI instructions. It is for fair comparison with the datasets that contain instructions without any examples, such as Alpaca, Inst. GPT-4 and Dolly. The original SNI instructions contain multiple positive and negative examples which further help the model better understand the task definition and expected output format. Because of the setting difference, the performance is much higher than what we report. In Table 4 of the original SNI paper, the intersection of row “Def” and column ”Def” shows the performance under the same input formulation as ours. The performance is 45.0, which is close to our reproduced results. Also, as shown in Line 334-337, we only use 90% of the original SNI training data and leave the other 10% as validation. The smaller training data size also contributes to the performance difference.
>
> [3] Wang et al., 2023, Self-Instruct: Aligning Language Models with Self-Generated Instructions, ACL.
>
> #### **Any references for BigScience and Flax Stack Exchange?**
> For BigScience, we leverage the datasets from the Huggingface spaces  https://huggingface.co/bigscience-data and [4]. For Flax Stack Exchange, we leverage the resource from [5]. (https://huggingface.co/datasets/flax-sentence-embeddings) We will add the references in our submission paper.
>
> [4] BLOOM, 2022, Huggingface BigScience Workshop, https://huggingface.co/bigscience/bloom.
> [5] Flax Sentence Embeddings Team, 2021, Stack Exchange Question Pairs.
>
> #### **How many instances were used for human evaluation?**
> The number of instances in every human evaluation for the comparison between two model outputs is 252. It is the full size of the User-Instruction-252 dataset. For each of the 252 instances, there are 3 different annotators to score the comparison between two model outputs to mitigate the randomness.

---

### Meta-Review · Area_Chair_mE6U · 2023-09-18

**Recommendation:** 4

**Metareview:**

Describes a method to generate instruction-tuning datasets from NLP datasets (e.g. on HuggingFace) in a more automatic fashion than prior work. In particular, the creation of templates for transforming inputs to instruction examples that are normally done by humans (e.g. PromptSource) is done by an LLM. The resulting large dataset (800k examples) is Dynosaur and is produced inexpensively ($12 querying ChatGPT). Base LLMs are fine-tuned on Dinosaur, and performance on Super-NI (ROUGE-L, automatic) and human evaluation are measured and outperforms competitors such as Alpaca.

The paper provides a cheap, and effective tool in the instruction-tuning dataset generation tool-box.

---

### Decision · Program_Chairs · 2023-10-07

**Decision:**

Accept-Main

**Comment:**

Describes a method to generate instruction-tuning datasets from NLP datasets (e.g. on HuggingFace) in a more automatic fashion than prior work. In particular, the creation of templates for transforming inputs to instruction examples that are normally done by humans (e.g. PromptSource) is done by an LLM. The resulting large dataset (800k examples) is Dynosaur and is produced inexpensively ($12 querying ChatGPT). Base LLMs are fine-tuned on Dinosaur, and performance on Super-NI (ROUGE-L, automatic) and human evaluation are measured and outperforms competitors such as Alpaca.

The paper provides a cheap, and effective tool in the instruction-tuning dataset generation tool-box.